# *Rhodotorula kratochvilovae* CCY 20-2-26—The Source of Multifunctional Metabolites

**DOI:** 10.3390/microorganisms9061280

**Published:** 2021-06-11

**Authors:** Dana Byrtusová, Martin Szotkowski, Klára Kurowska, Volha Shapaval, Ivana Márová

**Affiliations:** 1Faculty of Science and Technology, Norwegian University of Life Sciences, P.O. Box 5003, 1432 Ås, Norway; dana.byrtusova@nmbu.no (D.B.); volha.shapaval@nmbu.no (V.S.); 2Faculty of Chemistry, Brno University of Technology, Purkyňova 464/118, 612 00 Brno, Czech Republic; xcszotkowski@fch.vut.cz (M.S.); Winklerova10@gmail.com (K.K.)

**Keywords:** *Rhodotorula kratochvilovae*, lipids, extracellular glycolipids, carotenoids, β-glucan

## Abstract

Multifunctional biomass is able to provide more than one valuable product, and thus, it is attractive in the field of microbial biotechnology due to its economic feasibility. Carotenogenic yeasts are effective microbial factories for the biosynthesis of a broad spectrum of biomolecules that can be used in the food and feed industry and the pharmaceutical industry, as well as a source of biofuels. In the study, we examined the effect of different nitrogen sources, carbon sources and CN ratios on the co-production of intracellular lipids, carotenoids, β–glucans and extracellular glycolipids. Yeast strain *R. kratochvilovae* CCY 20-2-26 was identified as the best co-producer of lipids (66.7 ± 1.5% of DCW), exoglycolipids (2.42 ± 0.08 g/L), β-glucan (11.33 ± 1.34% of DCW) and carotenoids (1.35 ± 0.11 mg/g), with a biomass content of 15.2 ± 0.8 g/L, by using the synthetic medium with potassium nitrate and mannose as a carbon source. It was shown that an increased C/N ratio positively affected the biomass yield and production of lipids and β-glucans.

## 1. Introduction

Multifunctional biomass, i.e., the possibility of isolation of a broad spectrum of metabolites from one single fermentation process, is the perspective task that establishes a cost-effective microbial production and ensures the economic feasibility of microorganism cultures. The increasing demand for bio-based compounds has resulted in intensive screening of potential co-production candidates [1,2]. However, it is not easy to achieve high yields of all desired metabolites and cell biomass, since they require different culture strategies, and therefore, it is challenging to develop a balance between the medium composition and the suitable biomass and production of desired compounds. The co-production strategy focuses mostly on two-component yields, such as exopolysaccharides and lipids [2], carotenoids and lipids biosynthesis [3] or carotenoids and enzymes [4], of yeast.

Basidiomycota, the phylum of the fungus kingdom, contains 30,000 described species, which includes mushroom-forming fungi, jelly fungi, yeast, rusts and smuts [5]. Among the mentioned phylum, there is carotenoids-producing yeast (called “carotenogenic” or “red” yeast), in which the genus of *Rhodotorula* is the most studied. This yeast can utilize a wide range of substrates, mono- and polysaccharides and waste materials while producing high-value substances [6]. Besides carotenoids, these unicellular microorganisms are capable of producing a variety of biotechnologically important metabolites, such as lipids [7], ergosterol [8], enzymes [9], extracellular molecules [10] and polysaccharides [11].

Carotenoids, the terpenoid pigments of 40 carbon atoms derived from two units of geranyl-geranyl transferase pyrophosphate, represent important bioactive molecules for human health [12]. They are lipid-soluble natural colorants, with a yellow to red color, that exhibit antioxidant activity and promote healthy effects, such as a reduction of the risk of degenerative diseases (heart diseases, cancer, cataract or macular degeneration) [13] and improving immunity [14], and they are also vitamin A precursors. Most of the carotenoids are extracted from vegetables, requiring seasonal and geographic variability, or are produced by chemical synthesis [12], generating hazardous wastes and by-products. Commercial microbial sources focus mainly on algal sources, such as *Dunaliella* species (producer of β-carotene), *Chlorella* (luteine) or *Haematococcus* (astaxanthin) [15]. On the other hand, carotenogenic yeast, mainly *Rhodotorula* genera, can synthesize specific carotenoids (thorulene, torularhodin, β-carotene) in different proportions and yields [16]. Currently, renewable energy has been gaining scientific interest due to increasing demands of fossil fuels and the rise of global warming and environmental pollution. Microbial systems, mainly carotenogenic oleaginous yeast, are efficient producers of triacylglycerols (TAGs) and are able to accumulate up to 70% of dry cell weight (DCW) together with high amount of biomass [17]. Additionally, after the extraction of lipophilic compounds from biomass, other valuable products can be obtained. Beta-glucans are a group of polysaccharides, consisting of glucose monomers connected with a β-glycosidic bond. They can be found in cell walls of high fungi (mushrooms), yeast, plants, algae and bacteria, where they are responsible for the mechanical and structural properties [18,19]. Their increasing market value is caused by its properties, such as immunostimulatory, antimicrobial, antitumor, anti-oxidative and anti-inflammatory activities [20]. For carotenogenic yeast, only a few papers can be found concerning β-glucans production [11,21]; thus, it is an attractive task for identifying new potential biotechnological candidates.

The aim of this study was to assess the optimization of the production of a wide range of metabolites, from pigments, ergosterol and lipids to polysaccharides in the carotenogenic yeast *Rhodotorula kratochvilovae* CCY 20-2-26. The main goal of the presented study is to optimize the co-producing strategy and find an well-balanced and cheap culture medium.

## 2. Materials and Methods

### 2.1. Experimental Design

Two biological replicates were prepared for each strain and cultivation medium by inoculating strains from frozen cryopreserved stocks on agar plates. Every biological replicate was prepared in a separate Erlenmeyer flask. At the diverse time points of cultivation, cell biomass was examined by gas chromatography for lipid analysis, liquid chromatography for carotenoids and ergosterol production and glucan analysis by the Yeast and Mushroom β-glucan Assay Procedure. The extracellular component was extracted by ethyl acetate.

The experimental part starts with optimizing the culture conditions to enhance the biomass and multivariate metabolite production (nitrogen sources, carbon sources and the C/N ratios).

### 2.2. Yeast Strain and Media for Optimizing the Culture Conditions

Carotenogenic *Basidiomycetes* yeast strain *Rhodotorula kratochvilovae* CCY 20-2-26 was obtained from the Culture Collection of Yeasts (Institute of Chemistry, Slovak Academy of Science, Bratislava, Slovakia). Cultivation of yeast was performed, first, on YPD agar medium (yeast extract, 10.0 g/L; peptone, 20.0 g/L; glucose 20.0 g/L; agar, 20.0 g/L) (Merck, Darmstadt, Germany) inoculated from frozen cryopreserved stock cultures and incubated for 72 h at 25 °C. Inoculum was prepared by transferring 1 µL of yeasts cells from YPD agar into 50 mL of sterile YPD broth medium (yeast extract, 10.0 g/L; peptone, 20.0 g/L; glucose 20.0 g/L) (Merck, Darmstadt, Germany) in an Erlenmeyer flask (250 mL) and cultivated for 24 h at 25 °C under shaking regime (110 rpm, 50 mm). The cells were washed with sterile water and resuspended in fresh production medium in a volume ratio of 1:5 of YPD inoculum to production media (Table 1 and Table 2). The cultivation in production media was performed in Erlenmeyer flasks at 25 °C under the constant shaking regime (110 rpm, 50 mm) and the samples were taken every 48, 72, 96 and 144 h. All cultivation media were sterilized at 121 °C for 15 min.

### 2.3. Preparation of Yeast Biomass for the Glucan, Lipid and Carotenoids Analysis

After cultivation, the yeast biomass was centrifuged at 4500 rpm for 5 min at 4 °C and the biomass pellet was then washed three times using 0.1% NaCl solution. Furthermore, yeast biomass was freeze-dried for 48 h and, subsequently, stored at −20 °C until use.

### 2.4. Analysis of Glucans by Yeast and Mushroom β-glucan Assay Kit

The total glucan content and the content of β- and α-glucans were determined according to the Yeast and Mushroom β-glucan Assay Procedure K-YBGL (Megazyme/NEOGEN, Lansing, MI, USA) [22,23]. To estimate the total glucan content, freeze-dried yeast biomass was hydrolyzed with ice-cold 12 M sulfuric acid for 2 h and then incubated for 2 h at 100 °C. Furthermore, acidic hydrolysate was neutralized with 200 mM sodium acetate buffer (pH 5) and 10 M KOH, followed by the effect of enzymes exo-β-(1→3)-D-glucanase and β-glucosidase in acetate buffer (pH 4.5) for 1 h at 40 °C. These enzymes are able to hydrolyze extremely resistant beta-glycosidic bounds in glucans. The α-glucan content was determined after enzymatic hydrolysis with amyloglucosidase and invertase, which hydrolyze alpha-glycosidic bonds. The β-glucans content was calculated according to the recommendation of the assay kit procedure as the difference between total glucan and α-glucan content. The absorbance values indicating the total glucans and α-glucans content were obtained spectrophotometrically at 510 nm after adding glucose oxidase/peroxidase reagent.

### 2.5. Total Lipid Content and Analysis of Fatty Acid Profile

Into a 2 mL glass crimp vial, 10 ± 2 mg of freeze-dried yeast biomass was weighed together with 1.8 mL of transesterification solvent (15% (*v*/*v*) H_2_SO_4_ in methanol (HPLC grade), 0.5 mg/mL C17 as an internal standard) and the sample was incubated for 2 h at 85 °C. After cooling to room temperature, the full volume of vial was transferred into a 4 mL glass vial with the addition of 0.5 mL 0.05 M NaOH and 1 mL of hexane (GC grade). The mixture was vortexed for 5 min, and after the phase separation, 0.1 mL upper hexane extract and 0.9 mL of pure hexane was transferred into glass vials for further GC analysis. Total lipid content (wt% of total FAMEs of the dry weight) and the fatty acid profile were performed by Thermo Scientific TRACE™ 1300 Gas Chromatograph equipped with a Zebron ZB-FAME column, 30 m × 0.25 mm × 0.20 µm and flame ionization detector (FID). The temperature gradient is presented in the Table 3. In total, 1 μL of the sample was injected in split mode with an inlet temperature of 260 °C. FAME standard mixture (C4—C24; Sigma Aldrich/Merck, Darmstadt, Germany) dissolved in hexane was used for the identification of the FAMEs. Quantification was based on the C17:0 internal standard and relative response factors (RRF) calculated from five-point calibration curves of the individual FAMEs present in the standard mixture.

### 2.6. Analysis of Carotenoids

The method for the isolation and analysis of carotenoid pigments and ergosterol was adapted from Szotkowski et. al (2019) [24]. Briefly, 15 ± 3 mg of freeze-dried biomass was weighed into plastic extraction tubes and rehydrated by the addition of 1 mL of distilled water. The water was removed by centrifugation (10,000 rpm/5 min/10 °C), and to the pellet, 300 ± 20 mg of acid-washed glass beads (250–500 μm diameter, Roth, Germany) and 1 mL of methanol was added. The rupture of the biomass was performed by 10 min of vortexing on the bench-top vortex (2500 rpm). The content of the PP tube was transferred into a glass reaction tube by washing it with a 2000 μL of chloroform and the glass tube was vortexed for another 10 min; then, 1 mL of distilled water was added for the phase separation. After centrifugation (3000 rpm/5 min/4 °C), the separated bottom chlorophorm phase with extracted pigments and ergosterol was evaporated under nitrogen at 25 °C, followed by the addition of 1 mL of a mixture of ethylacetate:acetonitrile (20:60). The ethylacetate:acetonitrile mixture, containing extracted pigments, was filtered through a syringe filter (0.45 µm, PTFE membrane, 13 mm) and transferred into glass vials for further HPLC analysis.

The mobile phases consist of mixture A (84% of acetonitrile, 2% of methanol and 14% 0.1 M Tris-HCl (pH = 8) and mixture B (68% of methanol and 32% of ethylacetate). The conditions of separation are presented in Table 4. The contents of individual pigments (beta-carotene, lycopene, torulene, torularhodin) were calculated using calibration standards according to [24]. Thermo Finnigan Surveyor HPLC/PDA system (Thermo Fisher Scientific, Waltham, MA, USA) and Xcalibur software was used for chromatography data analysis. Beta-carotene and lycopene were purchased from Sigma Aldrich/Merck, Darmstadt, Germany, torularhodin and torulene from Carote*Nature*, Ltd., Münsingen Switzerland. 

### 2.7. Isolation and Characterization of Extracellular Glycolipids 

Exoglycolipids were isolated from culture medium according to the procedure from Wang et al. (2019) [25]. After cultivation, 20 mL of culture broth was centrifuged (6500 rpm/5 min) and the supernatant was separated and washed with ethylacetate in two steps (5 mL for each step). The exoglycolipids contained biomass was washed twice with 5 mL of ethylacetate and centrifuged (6500 rpm/5 min). All ethylacetate fractions were merged together, evaporated under nitrogen flow and lyophilized. The dry weight of the extract was weighed to estimate the grams of extracellular glycolipids per liter of culture medium. The final crude product was stored at 4 °C for further analysis. 

Purified exoglycolipids fraction was then analyzed for lipid and sugar content. The lipid content and composition were analyzed according the procedure in Section 2.7. For sugar content, 100 mg of crude product was put into a 10 mL glass vial with a silicone screw cap, together with 3 mL of 1% H_2_SO_4_ and heated to 100 °C for 2 h. After hydrolysis, sample was neutralized with 5% NaOH solution. The sample was filtered via a 0.4 μm filter into a 1.8 mL screw cap vial. The prepared sample was analyzed on a DIonex UltiMate 3000 series HPLC with an RI detector on Luna Omega Sugar Column (200 mm × 2.6 μm × 5.0 mm) using isocratic elution 75:25 ACN:MiliQ water at 30 °C for 30 min according to [24]. 

## 3. Results

### 3.1. Biomass and Biochemical Profile of R. kratochvilovae CCY 20-2-26 Growth on Diverse Nitrogen Sources

First, the optimization of cultivation conditions and medium composition was tested using various nitrogen and carbon sources. In the starting experiment, *R. kratochvilovae* CCY 20-2-26 grew in the presence of various nitrogen sources, such as potassium nitrate, yeast extract, ammonium sulfate, ammonium chloride and urea (see Table 1). Here, glucose was used as a sole carbon source at a C/N ratio of 70:1. Figure 1 shows the biomass production through time at different nitrogen sources. Potassium nitrate, yeast extract and urea appear to be the most effective for the growth of *R. kratochvilovae* CCY 20-2-26, where the yields reached a value of about 9.0 g/L of biomass. The highest biomass was obtained using potassium nitrate at the 144 h of cultivation, namely 9.5 g/L. On the contrary, there is only a slightly increase in biomass when ammonium sulfate and ammonium chloride were used. Therefore, these sources were no longer used in the following experiments. In general, the highest biomass production was recorded at 144 h, when the yeast reached the stationary phase.

Changes of glucose concentration in medium with different N sources is also documented in Appendix A. The data confirmed that the highest degree of glucose utilization by *R. kratochvilovae* CCY 20-2-26 cells occurred in the presence of urea and potassium nitrate as N sources. These compounds were used in the following experiments, where various simple sugars were tested as carbon sources (see Section 3.2). 

The accumulation of metabolites was affected by changing the nitrogen sources. Potassium nitrate, yeast extract and urea showed a decrease of the beta-glucan yield through time. Conversely, the ammonium sulfate reached maximum values at the end of cultivation, 17.60 ± 1.84% *w*/*w* (Table 5). In Appendix A more detailed data are introduced regarding the production of alpha-glucans and beta-glucans. These data show that the production of alpha-glucans is very low and, in most of the cultivations, do not exceed 1–2%. Thus, the total intracellular glucans can be considered as the beta-glucans with a small portion of alpha-glucans.

Intracellular lipids increased to almost 60% of the DCW at 96th with potassium nitrate, followed with yeast extract and urea. The most abundant fatty acids were oleic acid (46.6%), palmitic acid (18.3%), linoleic acid (18.6%), stearic acid (4.5%), α-linolenic and myristic acid (3.1 and 2.1%). The production of individual fatty acids on different N sources can be found in a more detailed form in Appendix A. These data confirmed that nitrogen sources had only a marginal influence on the fatty acid composition in the intracellular lipids of *R. kratochvilovae* CCY 20-2-26. 

The biosynthesis of exoglycolipids showed the same trend as at intracellular lipids, where maximum values reached up to 1.8 g/L (Figure 2, Table 5). The lowest yield of extracellular lipids was observed at 72 and 144 h of cultivation using ammonium sulfate, i.e., only 0.1 g/L. The composition of the glycolipid showed 44% lipid content, composed mainly from oleic acid (47.7%), linoleic acid (22%), palmitic acid (11.8%), linolenic acid (10%) and a minority of stearic and capric acid (4.9 and 2.3%). The fatty acids from glycolipid showed a similar composition to intracellular lipid, but with a higher content of palmitic acid. 

The carotenoids content varies from 0.97–2.83 mg/g of biomass. Potassium nitrate and urea increased the carotenoids production to over 2 mg/g of DCW at the end of cultivation. The optimal time point for maximum accumulation of intracellular lipids is 96 h (Table 5), while prolonged growth (144 h) can be seen more suitable for pigments and extracellular glycolipids production. Detailed distributions of individual carotenoid pigments in media with different N sources are documented in Appendix A. There are only small differences in the ratio of the main produced pigments—beta-carotene and more oxidized torulene + torularhodine in cells grown in the presence of different N sources. 

### 3.2. Biochemical Profile of R. kratochvilovae CCY 20-2-26 Growth on Diverse Carbon Sources

From previous experiments, potassium nitrate and urea were chosen as the best nitrogen sources. In further experiments, the consumption of different sugars (lactose, mannose, glycerol, xylose and glucose) as C sources were compared during cultivation in the presence of and potassium nitrate (Figure 3). These data confirmed glucose as the most preferred substrate for *R. kratochvilovae* CCY 20-2-26. However, in the medium with potassium nitrate as an N source, mannose was utilized by very similar kinetics as glucose (Figure 3B). Because of the comparable growth and biomass production of yeast cells on mannose and urea (Appendix A) and based on recently published data [11,24], only mannose was used in further experiments and compared with other sugars. The reason is that all these tested sugars could be obtained by hydrolysis of some waste by-products as a rest material. 

These nitrogen sources are cheap and simple for media preparation; moreover, they showed the highest yields of biomass, intracellular lipids and mainly exoglycolipids (Figure 1, Table 5). High yields were also observed for the yeast extract, which is, however, a demanding source due to its price compared to the two selected above. Regarding metabolite production, mannose was found to be a very efficient carbon source. Biomass production showed values of about 6.0 g/L at the beginning of the experiment, and up to 10.0 g/L of biomass after 144 h of cultivation. Thus, urea and mannose appear to be the best sources of carbon and nitrogen for biomass production over time. In the previous experiment, using urea as a nitrogen source, a maximum yield of about 9.0 g/L was obtained, when the carbon source was glucose. Here, it can be observed that even better results (up to 10.0 g/L) were obtained with the same nitrogen and mannose source as the carbon source (Figure 4). Thus, it seems that mannose is an even better source of carbon than glucose for this experiment, although the difference is not significant. The worst source in the experiment was lactose, which is the least suitable source of carbon for yeast, probably due to the absence of lactase to break down disaccharide into usable monosaccharides. The biomass yield was maximally only about 1.0–1.5 g/L. Lactose was followed, also as a less suitable carbon source, by glycerol, which showed values slightly above 2.0 g/L of biomass, and these values were very similar throughout the whole time horizon of cultivation. Xylose appears very similar to glycerol, which showed slightly better results with 3.0 g/L of biomass. Corresponding data regarding utilization of individual carbon sources are shown in Appendix A.

Table 6 shows the production of intracellular lipids, carotenoids, extracellular glycolipids and β-glucan in yeast biomass cultured at different saccharides and glycerol with a combination of urea. The biomass content for lactose was too low for glucan measurements. The urea with combination of mannose, glycerol and xylose decreases the β-glucan content when compared to previous experiment with glucose. With mannose as a carbon source, glucan yield dropped from 13.75 ± 0.99 to 10.54 ± 0.76, probably due to the accumulation of other metabolites. Conversely, cultivation on glycerol and xylose showed a slight increase in β-glucan production at the end of cultivation. In Table 6, values of the total intracellular glucans are introduced, while detailed production of alpha-glucans and beta-glucans in media with different C sources and urea as an N source is illustrated in Appendix A.

The production of exoglycolipids, using different carbon sources, is analogical to the production of biomass—the most suitable carbon source is mannose (1.4 ± 0.10 g/L). The next suitable source appears to be glycerol, where the maximum yield was reached after 96 h of cultivation, slightly above 0.2 g/L of exoglycolipids. Other sources showed very low values compared to mannose, around 0.1 g/L. The concentration of exoglycolipids was slightly lower in this experiment when compared to the previous experiment. Thus, glucose in combination with urea appears to be a more suitable source of the production of exoglycolipids. 

Regarding intracellular lipids, mannose was confirmed to be the most suitable carbon source. However, not so high values were achieved as in the cultivation with potassium nitrate and glucose (Table 5). Using mannose, the maximum value was reached at 96 h of cultivation (about 55% of the intracellular lipids in DCW). At 144 h, the lipid yield dropped to 51%, probably because of the exhaustion of carbon source. For glycerol and xylose, the yields were relatively similar. In media with glycerol, a maximum was reached at 72 and 144 h, with the production of approximately 17% of the intracellular lipid content. Xylose reached a maximum of 20% lipids of DCW at 144 h. The less suitable source here was lactose, which overall did not lead to the production of more than 8% of lipids. 

In the cultivations with potassium nitrate as a C source, the best source of carbon was mannose, particularly at 144 h, when the yield reached up to about 9.5 g/L of biomass (Table 7). From the beginning of the cultivation, the yield increased rapidly. This fact indicates that mannose is a suitable source of carbon for *R. kratochvilovae* CCY 20-2-26, also in combination with KNO_3_, when a short cultivation (48 h) resulted in a yield of 7.2 g/L biomass after 48 h. Data of utilization of individual sugars by *R. kratochvilovae* during growth on various carbon sources are shown in Figure 3 and Appendix A. 

In general, potassium nitrate is a less suitable source of nitrogen than urea, because, in the previous experiment, mannose values reached up to about 10.0 g/L, which is the maximum for biomass across all experiments performed. However, comparable biomass values, i.e., around 9.5 g/L, were obtained in media with glucose as a C source and the already mentioned potassium nitrate as an N source. 

As relatively suitable carbon source xylose was identified, which reached values of about 6.2 g/L at the end of the cultivation. This finding is very important because of the potential utilization of lignocellulose substrates. When comparing the amount of biomass in media with urea, in the presence of KNO_3_, about 2-times higher biomass production than for the combination of urea and xylose was found. Using mannose as a C source, it can be concluded that the combination with urea shows better results than with potassium nitrate, but on the contrary, xylose, as a carbon source, works better in combination with potassium nitrate than with urea. The worst carbon sources here are glycerol and especially lactose. While glycerol reached at least 3.38 g/L during cultivation, lactose showed much worse results, with a maximum value of 0.61 g/L. 

Table 7 shows the production of β-glucan in yeast biomass cultured at different carbon sources combined with potassium nitrate as a nitrogen source. Again, the biomass content for lactose was too low for glucan measurements. When compared with urea, a rapid increase in β-glucan yield occurred when cultured on mannose. At 144 h of cultivation, high β-glucan content was accompanied with biomass production of 9.42 g/L. On the contrary, glycerol shows the production of only 2.69 ± 0.34% of DCW. Detailed production of alpha-glucans and beta-glucans in media with different C sources and potassium nitrate as an N source is illustrated in Appendix A. 

The number of biosynthesized exoglycolipids is highest in medium with mannose, in which the highest yields (up to about 2.0 g/L) were achieved within 144 h of cultivation (Figure 5).

Compared with the cultivations on urea, potassium nitrate appears to be a more suitable source for exoglycolipid biosynthesis, given that in the previous experiment the maximum values were only about 1.4 g/L exoglycolipids. For lactose and glycerol, the exoglycolipids production did not exceed the value of 0.2 g/L. 

The highest percentage of intracellular lipids in dry matter, namely 57%, was accumulated in mannose medium with potassium nitrate as an N source. The other reliable carbon source was xylose, which reached about 35% within the 144 h of cultivation. The least suitable source was, again, lactose (20% of intracellular lipids in DCW). It can be stated that for the combination of glucose and potassium nitrate, the maximum value was reached within 96 h, namely 60% of lipids, but the production of exoglycolipids was lower compared to mannose.

The distribution of individual fatty acids in the total intracellular lipids is introduced in Appendix A for urea and Appendix A for potassium nitrate as an N source. There are no substantial differences between the used N sources, and the FA profiles are quite similar in the media with individual sugars. The distribution of FA depends on the type of sugar and the FA profiles in lipids produced by yeast cells grown on lactose and glycerol are different from other media. While the highest amount of oleic acid was produced in medium with lactose as C source, in glycerol media intracellular lipids with the highest fraction of saturated fatty acids were produced. While the highest amount of oleic acid was produced in medium with lactose as C source, in glycerol media intracellular lipids with the highest fraction of saturated fatty acids were produced.

The production of total carotenoid pigments is very low in media with lactose as a C source; in the rest of the media with monosaccharides and glycerol, the yields are very similar. The production of individual pigments in the media with urea and potassium nitrate are introduced in Appendix A, respectively. The pigment profiles in yeasts are similar to previous results (Appendix A), with a majority of beta-carotene, torulene and torularhodin. Some differences between the ratio of beta-carotene and more oxidized torulenes were observed, depending on the type of sugar in the cultivation medium. The highest portion of saturated fatty acids was found in glycerol and glucose media.

### 3.3. Influence of Different Carbon: Nitrogen Ratio on Growth and Metabolism of Rhodotorula kratochvilovae CCY 20-2-26

The C/N ratio is a very important criterion for achieving maximum production of selected metabolites. Regarding its influence on biomass production, the higher the C/N ratio, the higher the biomass yields we are able to achieve, mainly due to the accumulation of specific biomolecules [26]. Figure 6 describes the dependence of biomass production at different concentrations of mannose as a carbon source. Potassium nitrate was chosen as an N source with the C/N ratios of 20:1, 40:1, 100:1 and 120:1 (C/N 70:1 was used in previous experiments). The highest biomass yields were achieved at the highest concentration of mannose after 192 h of cultivation (15.2 ± 0.8 g/L). The situation was relatively favorable even with a C/N ratio of 100:1, where the highest yield was achieved after 168 h of cultivation, with a value exceeding 14.0 g/L of biomass. For C/N ratios of 20:1 and 40:1, the yields were significantly lower. As for a ratio of 40:1, the highest yield was obtained at 144 h of cultivation, after which the value rather decreased.

Table 8 shows the data of the production of metabolites in yeast biomass cultured at different C/N ratios, where a C/N ratio of 40:1 shows the highest β-glucan yield, 12.83 ± 1.23% of DCW. Taken together, at the C/N ratio of 70:1 (Table 7), the best results for β-glucan production were found, while better yields of biomass, intracellular lipids and exoglycolipids were observed at the C/N ratio of 120:1. Thus, higher C/N ratio is better for the majority of biotechnological applications. As mentioned previously, cultures that are deficient in nitrogen intensively produce metabolites, such as exoglycolipids. Therefore, the difference between low and high C/N ratios is obvious, the production is up to 12 times higher at a C/N ratio of 120:1 than at a ratio of 20:1. The highest value of exoglycolipids for a C/N ratio of 120:1 was reached at 168 h of cultivation. Very similar results can be obtained with a C/N ratio of 100:1. The production was many times lower for the C/N ratios 20:1 and 40:1, where the minimum value of the whole experiment was reached at 96 h (C/N 20:1), namely 0.05 g/L. Compared to previous experiments, the exoglycolipids production at a C/N ratio of 100–120 was higher than at a C/N ratio of 70:1. Thus, the C/N ratio played an important role in the production of exoglycolipids, especially regarding their increase.

After previous experiments focused on the selection of the most suitable sources of carbon and nitrogen, the C/N ratio was optimized to achieve the highest values of lipid yield. For high C/N ratios, values close to 70% of lipid accumulation were achieved. Particularly for the C/N 120:1 ratio, 67% of intracellular lipids in DCW was achieved after 168 h of cultivation. For the C/N 100: 1 ratio, the highest accumulation was observed after 168 h of cultivation (62%) as well, while for lower C/N ratios, the maximum value was reached at 96 h (40% for C/N 40: 1). The intracellular lipids composition (at C/N 100) was represented mainly by oleic acid (29.7%), palmitic acid (20.2 0%), linoleic acid (27.1%), stearic acid (3.1%), α-linolenic and myristic acid (7.4 and 2.1%). In the previous experiment, the maximum of the total lipids (60%) was reached at 144 h of cultivation (C/N ratio of 70:1). In the experiment with mannose and urea, the maximum value of lipids was 55%, which is a number slightly lower than in the experiment with potassium nitrate.

The fatty acid composition of intracellular lipids produced at different C/N ratios during 96 h of cultivation on mannose and potassium nitrate is introduced in Appendix A. The data show that with an increased C/N ratio, the production of saturated fatty acids increased, while the production of PUFA (linoleic acid C18:2, linolenic acid C18:3) gradually decreased. The production of total carotenoids exhibited no substantial changes with an increased C/N ratio, while the distribution of pigments was different. An increased C/N ratio led to gradually decreased production of beta-carotene accompanied by increased production of oxidized torulenes (Appendix A). These changes could be connected with increased intracellular stress (particularly oxidative).

## 4. Discussion

Carotenogenic yeasts are exceptional microorganisms able to biosynthesize a broad range of high-valuable metabolites, such as lipids, pigments, polysaccharides and enzymes. With such a versatile metabolism, they can establish the economic feasibility of microbial cultures by choosing the co-production strain. The production and study of the metabolites of oleaginous carotenogenic yeasts represents a major challenge for industrial biotechnology [11,24,27,28]. Red yeasts could be directly used in the form of high-value multifunctional biomass in feed and food nutrition, pharmacology and medicine. 

Red yeasts can utilize a wide range of growth substrates, including by-products from the agriculture supply chain. When sugars or related substances (glycerol) are used as substrates for yeast cultivation, lipid accumulation is initiated after the exhaustion of nitrogen from the medium. Oleaginous fungi are able to produce and accumulate up to 85% (*w*/*w*) of lipids, consisting mainly of triacylglycerols (TAGs). Nitrogen depletion leads to a rapid decrease in the concentration of cellular AMP, which is further metabolized to complete nitrogen needs. A decrease of AMP concentration alters the Krebs cycle, resulting in the accumulation of citric acid, which is gradually secreted from mitochondria to the cytoplasm by tri-carboxylate transport system. In cytosol, citric acid is cleaved by ATP-citrate lyase (characteristic enzyme for oleaginous microorganisms) into acetyl-CoA and oxaloacetate. After activation of acetyl-CoA carboxylase by citric acid, acetyl-CoA generates intracellular fatty acids and subsequently triacylglycerols (TAGs) [28,29]. In non-lipid producing microorganisms, nitrogen exhaustion led to a transport of citric acid into the growth medium. Citric acid can also inhibit key control enzymes of glycolysis (phospho-fructokinase), leading to the inhibition of glucose catabolism and, consequently, to the accumulation of intracellular polysaccharides [11,28,29]. 

In the present study, the influence of nitrogen sources (yeast extract, potassium nitrate, ammonium chloride and urea), carbon sources (glucose, mannose, xylose, lactose and glycerol) and CN ratios (20:1, 40:1, 70:1, 100:1 and 120:1) were studied as the main features that can modify the metabolite accumulation. For the evaluation of the co-production strategy, red yeast *R. kratochvilovae* CCY 20-2-26 was chosen due to its high intracellular lipid and extracellular glycolipid production, as published elsewhere [27]. This strain was partly studied in our recent study [11], in which this strain was able to accumulate 38.21% (*w*/*w*) of lipids and 20.73% of β-glucans, accompanied with a high biomass yield (15.19 g/L). Biomass yield and the production of lipids and β-glucans were influenced by the C/N ratio and extracellular osmolarity. An increase in the C/N ration led to an increase in biomass, lipid and β-glucans production for several yeast strains, while osmolarity had a negative effect on the biomass and lipid production but positively affected β-glucans production.

Some other studies have been performed that employed some oleaginous yeasts, e.g., *Cryptococcus curvatus* [28,29,30], *Yarrowia lipolytica* and *Rhodosporidium* sp. [31,32] for the production of lipids in different fermentation configurations and culture media. In non-conventional yeast strains belonging to the genera *Yarrowia* and *Rhodosporidium* grown on biodiesel-derived glycerol as sole substrate, biomass production (DCW = 18.1–27.3 g/L) containing high amount of lipids (30.3–54.5% in DCW) was obtained, depending on the strain and media composition. Extracellular metabolites, such as citric acid (16–20 g/L) and polyols (mannitol, erythritol and arabitol; 21–48 g/L), were produced as well. At the late growth phases, storage metabolites were re-consumed [31], similarly to other studies [11,28,29,30,31,32] as well as the present study. 

*Cryptoccus curvatus* grown under nitrogen-limited conditions, with lactose or sucrose as C sources, accumulated high quantities of intracellular total sugars (ITS) at the beginning of fermentation (up to 68% *w*/*w*), with ITS values progressively decreasing to 20%, *w*/*w*, at the end of the fermentation [29]. Similar results were obtained in this study. In most cases (except glycerol + Gly, Xyl–Table 6; glucose + AS—Table 5 and mannose + KNO_3_—Table 7), the kinetics of beta-glucan production (expressed as the percentage of DCW as a function of culture time) seems to present relatively elevated polysaccharide values at the very early stages of the fermentation (in many cases, these values are >20% *w*/*w*). These values drop as the fermentation proceeds simultaneously with the increase of the values of cellular lipid (% in DCW). This is further evidence for a typical inter-play in the biosynthesis of the cellular polysaccharides and cellular lipids that has been reported above [28,29,30,31,32].

In red yeasts, carotenoids are produced as another type of stress metabolite. Previously, some control mechanisms were described for the simultaneous production of lipids, pigments and intracellular beta-glucans in some red yeast strains [11,24]. The tested strains exhibited a slight decrease of the total glucan content with increased C/N ratios. It is well known that a high C/N ratio can have a positive effect on the accumulation of some intracellular metabolites (glucans, lipids, pigments, etc.). Thus, the increase in the biomass yield under a high C/N ratio could be a result of mainly higher accumulation of certain metabolites, but not because of the increase in cell proliferation. We have illustrated this fact by the introduction of the summary number of glucans and lipids accumulated in the biomass at a growing C/N ratio [24]. With the exception of the lowest C/N ratio, when the ratios of glucans and lipids were quite differed in individual strains, at a higher C/N ratio, we observed quite stable values of glucans as well as the typical sums of glucans and lipids for individual strains. The sum of glucans and lipids formed 50–60% of the biomass and changed only slightly with a further increase of the C/N ratio. Similar results were also found in this study (Table 8), where the sum of lipids and beta-glucans varied in a range of about 22–26% of CDW at a C/N ratio of 20:1 and increased to about 60–80% of CDW at a C/N ratio of 120:1. The increase of lipids at a higher C/N ratio was about 3–4×, while the concentration of accumulated beta-glucans changed slightly and did not exceed approximately 20%. 

The C/N ratio appeared to be the most effective parameter for enhancing the microbial lipid biosynthesis, as published elsewhere [26]. Due to the AMP-dependent isocitrate dehydrogenase enzyme, some yeasts are able to produce up to 70% lipids per DCW [26]. By optimizing the culture conditions, we were able to achieve 66.7 ± 1.5% of lipids per DCW (over 10 g/L) using potassium nitrogen as a nitrogen source and mannose as a carbon source at the C/N ratio 120:1. The intracellular lipids composition was mainly represented by oleic acid (29.7%), palmitic acid (27.2%), linoleic acid (21.1%), stearic acid (1.6%), α-linolenic and myristic acid (7.4 and 2.1%). Our results are one of the highest compared to lipid production per DCW and published elsewhere [26,27,33].

Some of the *Rhodotorula* yeast strains can synthesize extracellular products, such as glycolipids, termed polyol esters of fatty acids (PEFA). Garay et al. (2017) researched 65 yeast strains, of which 19 produced PEFA in the range of 1.1–12.4 g/L [34]. In our experiments, potassium nitrate, urea and yeast extract with a combination of glucose and mannose caused the highest increase in glycolipid secretion, namely 2.42 ± 0.08 g/L. Similar results were described in *R. babjevae* Y-SL7 cultured on glucose and yeast extract, but low on urea [35]. According to the literature, cultivation in bioreactors provided higher yields [34,35]. Garay et al. (2017) produced 20.9 ± 0.2 g/L and 11.2 ± 1.6 g/L of PEFA by *R. aff. paludigena* UCDFST 81-84 and *R. babjevae* UCDFST 04-877 using glucose and a mixture of ammonium chloride and yeast extract as a nitrogen source [36]. In another study, the yield increased to 48.5 g/L using inulin as a carbon source [25]. Thus far, the high production cost of biosurfactants is still the main barrier to its commercialization, the same as for microbial oils. The promising solution can be the biosynthesis of multiple metabolites and the use of low-cost media or waste substrates [37].

In isolated extracellular glycolipids, we also tried to analyze the composition of lipid and saccharide fractions; on average, the saccharide content makes up 45–57% (*w*/*w*) and the lipid content 43–55% (*w*/*w*). With regards to the extracellular lipid fraction, the fatty acid composition was similar to intracellular lipids with an average SFA:MUFA:PUFA ratio of 19:48:33 [24]. With regards to the saccharide fraction, the HPLC analysis proved the presence of glucose molecules in the sample. Based on our results, non-hydrolyzed saccharide is with high probability disaccharide sophorose. Due to a lack of standard, we have identified it according to the retention time of other disaccharides cellobiose and lactose, which elute at the same time. The difference between molecules is only in the position of a glycoside bond, which has minimal impact on the elution time and elution differences between disaccharides in this method. This presumption is supported by literature sources [24,36,37,38]. Due to different structures between the individual PEFA that are produced and low yield, we cannot with certainty calculate the number of glucose molecules/sophorose molecules in PEFA.

Today’s commercial production of β-glucans is focused mainly on *Saccharomyces cerevisiae* and higher fungi [22,23,39]. It represents the unexplored biotechnological potential within carotenogenic yeast, and only a limited number of publications can be found. Our previous study showed that the β-glucan content is strain-dependent and ranged from 9–32% per dry cell weight [11]. Here, the culture conditions, time and accumulation of other metabolites significantly affected the cell wall polysaccharide content. Different nitrogen sources did not affect the β-glucan content significantly (13–17% per DCW in later growth); carbon sources, on the other hand, showed high variability. The decrease in β-glucan production within the cultivation time can be explained by the accumulation of lipids, as described above. 

Carotenoids from biomass of the *Rhodotorula* genus can be classified as low (less than 0.1 mg/g), medium (0.1–0.5 mg/g) and high (more than 0.5 mg/g) [12]. The best results were achieved for urea and potassium nitrate with a combination of glucose, 2.83 ± 0.21 and 2.48 ± 0.13 mg/g of biomass, respectively. With other carbon sources, glycerol shows a presence of 2.72 ± 0.24 mg/g of total carotenoids, accompanied by 2.21 ± 0.16 mg/g mannose for a C/N ratio of 70:1. The composition of carotenoid pigments was relatively stable at most of the cultivation conditions, independently from the type of carbon and nitrogen source, respectively. Major fractions formed beta-carotene and a mixture of more oxidized torulenes (torulene and torularhodin). Both these derivatives have a similar biological effect and, thus, they were evaluated as a sum of the total carotenoids. The production of the total carotenoids exhibited no substantial changes also with an increased C/N ratio, while the distribution of pigments was different. An increased C/N ratio led to gradually decreased production of beta-carotene, accompanied by increased production of oxidized torulenes.

Taken together, this study provides insight into the co-production of four metabolites with the application of cheap synthetic medium. The studied strain, *R. kratochvilovae* CCY 20-2-26, represents an interesting candidate to study the simultaneous production of extracellular glycolipids, intracellular oils, glucans and carotenoids.

## 5. Conclusions

By optimizing the culture media, *R. kratochvilovae* CCY 20-2-26 was able to produce a high number of intracellular lipids, β-glucans and carotenoids, together with a high biomass yield. In addition, extracellular glycolipid, secreted into culture media, had a lipid content of 44% and was composed mainly of oleic acid (47.7%), linoleic acid (22%), palmitic acid (11.8%) and linolenic acid (10%). Using high C/N ratio with mannose as a carbon source and potassium nitrate as nitrogen source, 15.2 ± 0.8 g/L of yeast biomass was achieved, containing significant amount of intracellular lipids (66.7 ± 1.5% of DCW), exoglycolipids (2.42 ± 0.08 g/L), β-glucans (11.33 ± 1.34% of DCW) and carotenoids (1.35 ± 0.11 mg/g). Similar results were obtained with glucose as a carbon source. Conversely, lactose, xylose and glycerol declined the overall production. Potassium nitrate appeared to be an effective and cheap alternative nitrogen source compare to yeast extract. 

## Figures and Tables

**Figure 1 microorganisms-09-01280-f001:**
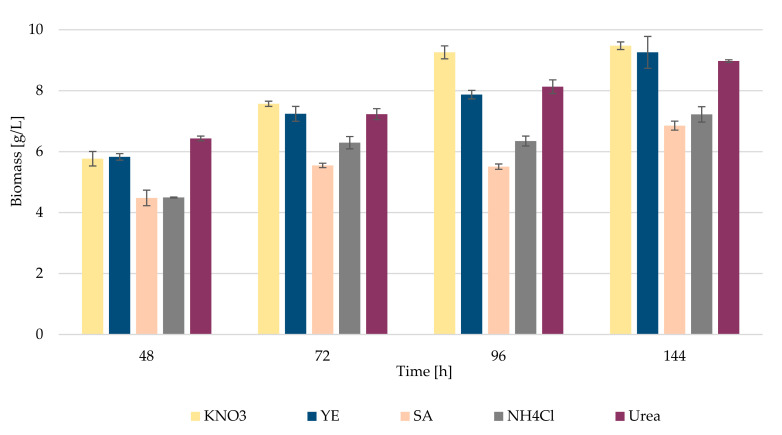
Biomass production by *R. kratochvilovae* CCY 20-2-26 cultivated in the presence of various nitrogen sources.

**Figure 2 microorganisms-09-01280-f002:**
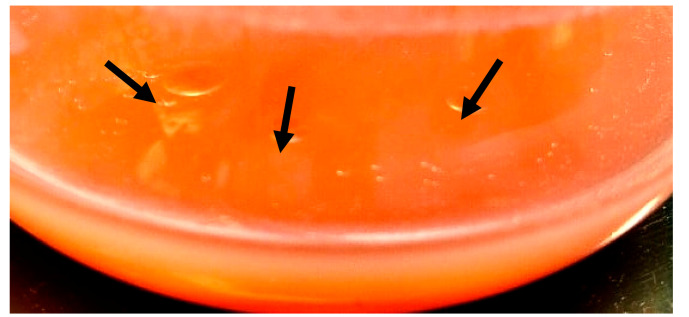
The presence of “oil droples” (black arrows) on the surface of culture medium of the yeast strain *Rhodotorula kratochvilovae* CCY 20-2-26.

**Figure 3 microorganisms-09-01280-f003:**
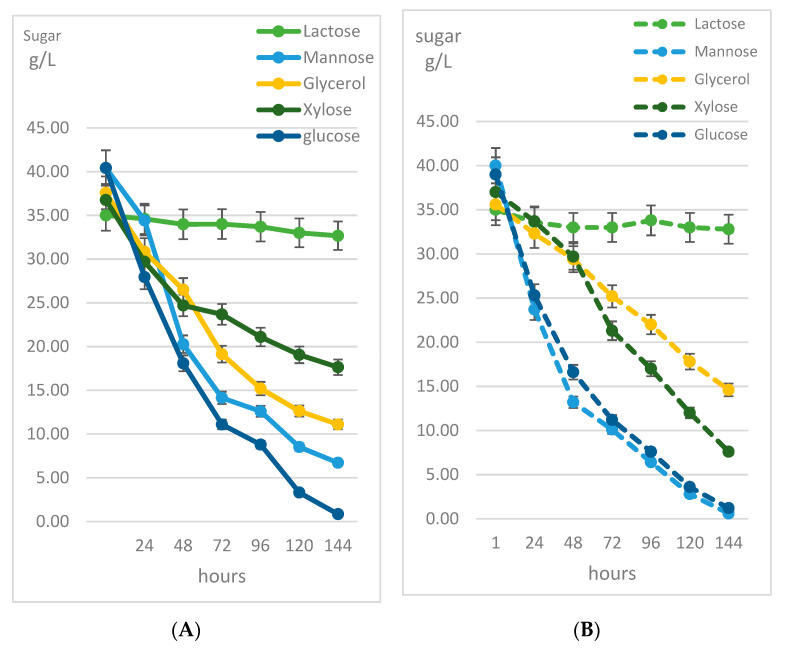
Consumption of different carbon sources by *R. kratochvilovae* CCY 20-2-26. (**A**) Changes of the concentration of individual sugars in the medium with urea as an N source. (**B**) Changes of the concentration of individual sugars throughout the cultivation in the medium with potassium nitrate as an N source.

**Figure 4 microorganisms-09-01280-f004:**
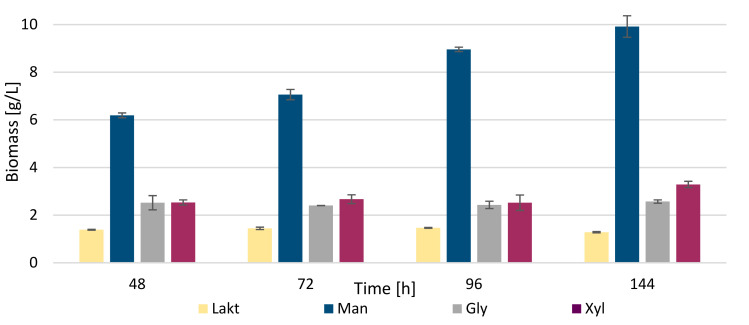
Biomass production with diverse carbon sources and with urea as nitrogen source. Abbreviations: Lakt: lactose; Man: mannose; Gly: glycerol; Xyl: xylose.

**Figure 5 microorganisms-09-01280-f005:**
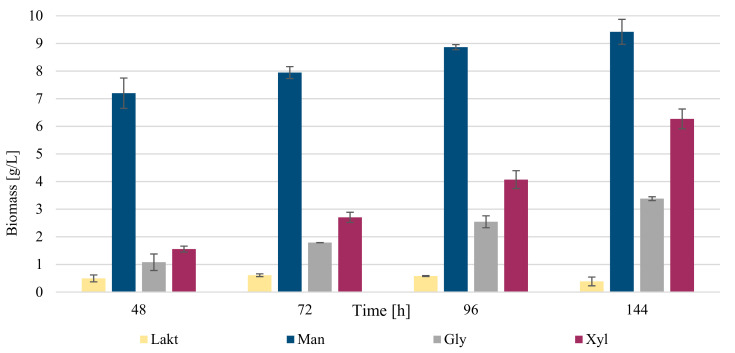
Biomass production with diverse carbon sources and with potassium nitrate as nitrogen source. Abbreviations: Lakt: lactose; Man: mannose; Gly: glycerol; Xyl: xylose.

**Figure 6 microorganisms-09-01280-f006:**
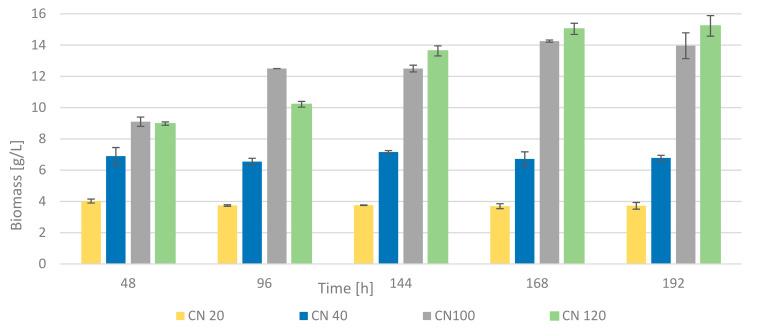
Biomass production with diverse C/N ratio and potassium nitrate as a N source.

**Table 1 microorganisms-09-01280-t001:** Composition of the culture media with different nitrogen sources (C/N ratio 70:1).

**Component**	**g/L**
Glucose	40.42
KH_2_PO_4_	4
MgSO_4_·7H_2_O	0.7
**Nitrogen sources**	**g/L**
Yeast extract	2
KNO_3_	1.52
NH_4_Cl	0.80
(NH_2_)_2_SO_4_	0.9906
Urea	0.45

**Table 2 microorganisms-09-01280-t002:** Composition of the culture media with different carbon sources (C/N ratio 70:1).

Component 1 [g/L]	Component 2 [g/L]
Urea—0.45	KNO_3_—1.52
KH_2_PO_4_—4	KH_2_PO_4_—4
MgSO_4_·7H_2_O—0.7	MgSO_4_·7H_2_O—0.7
**Carbon sources**	**g/L**
Mannose	40.42
Lactose	34.94
Glycerol	37.57
Xylose	36.75
Glucose	40.42

**Table 3 microorganisms-09-01280-t003:** Temperature gradient.

Retention Time (min)	Gradient (°C/min)	Target Temperature (°C)	Hold (min)
0	-	-	-
1	0	80	1
5	15	140	0
21.7	3	190	0
25.5	25	260	1
25.5	STOP	-	-

**Table 4 microorganisms-09-01280-t004:** Conditions of HPLC separation of carotenoids, sterols and ubiquinone.

Column	Kinetex, EVO 150 × 4.6 mm, 2.6 µm; Phenomenex
Volume of the sample	20 µL
Elution	Gradient 0–13 min: from 100% A to 100% B linearly 13–19 min: 100% B 19–20 min: from 100% B to 100% A linearly 20–25 min: 100% A
PDA	285 (ergosterol), 435, 450 and 680 nm (carotenoids)
Temperature	25 °C
Time of analysis	25 min

**Table 5 microorganisms-09-01280-t005:** Biomass composition of *R. kratochvilovae* CCY 20-2-26 cultured on different nitrogen sources.

Nitrogen	Metabolite	48 h	72 h	96 h	144 h
**KNO_3_**	Total lipids (% of DCW)	30.2 ± 1.9	42.5 ± 2.1	59.6 ± 1.3	49.8 ± 2.5
	Total carotenoids (mg/g)	1.00 ± 0.27	1.25 ± 0.13	1.36 ± 0.14	2.48 ± 0.13
	Extr. glycolipids (g/L)	0.62 ± 0.02	1.73 ± 0.09	1.51 ± 0.20	1.83 ± 0.07
	Beta-glucan (% of DCW)	22.05 ± 2.21	18.31 ± 1.83	15.24 ± 1.24	14.39 ± 0.77
**YE**	Total lipids (% of DCW)	37.0 ± 1.0	43.8 ± 1.8	58.8 ± 1.5	56.2 ± 1.2
	Total carotenoids (mg/g)	1.19 ± 0.11	1.08 ± 0.05	1.25 ± 0.03	1.16 ± 0.09
	Extr. glycolipids (g/L)	0.51 ± 0.04	0.75 ± 0.05	1.17 ± 0.20	1.28 ± 0.08
	Beta-glucan (% of DCW)	24.99 ± 1.69	21.43 ± 1.57	16.70 ± 1.33	12.62 ± 1.11
**SA**	Total lipids (% of DCW)	25.1 ± 2.0	34.6 ± 2.4	37.2 ± 0.5	27.0 ± 2.1
	Total carotenoids (mg/g)	1.30 ± 0.02	1.48 ± 0.16	1.61 ± 0.03	1.38 ± 0.06
	Extr. glycolipids (g/L)	0.12 ± 0.01	0.05 ± 0.02	0.08 ± 0.03	0.07 ± 0.02
	Beta-glucan (% of DCW)	12.41 ± 1.02	13.98 ± 1.04	16.54 ± 1.00	17.60 ± 1.84
**NH_4_Cl**	Total lipids (% of DCW)	30.5 ± 2.4	43.6 ± 2.1	41.3 ± 1.2	38.8 ± 3.0
	Total carotenoids (mg/g)	1.34 ± 0.10	1.54 ± 0.09	1.52 ± 0.16	1.52 ± 0.04
	Extr. glycolipids (g/L)	0.13 ± 0.01	0.31 ± 0.05	0.32 ± 0.09	0.23 ± 0.08
	Beta-glucan (% of DCW)	14.80 ± 0.81	15.71 ± 0.66	13.33 ± 0.85	11.30 ± 0.75
**Urea**	Total lipids (% of DCW)	31.3 ± 1.1	42.2 ± 2.0	56.8 ± 1.2	51.7 ± 3.1
	Total carotenoids (mg/g)	0.97 ± 0.01	1.04 ± 0.00	1.14 ± 0.46	2.83 ± 0.21
	Extr. glycolipids (g/L)	0.58 ± 0.02	0.88 ± 0.05	1.51 ± 0.07	1.57 ± 0.06
	Beta-glucan (% of DCW)	20.94 ± 1.25	20.15 ± 1.30	14.67 ± 0.59	16.93 ± 1.13

Abbreviations: YE: yeast extract; SA: ammonium sulphate.

**Table 6 microorganisms-09-01280-t006:** Biomass composition of *R. kratochvilovae* CCY 20-2-26 cultured on different carbon sources with urea.

Carbon	Metabolite	48 h	72 h	96 h	144 h
**Lactose**	Total lipids (% of DCW)	9.0 ± 2.3	8.6 ± 1.6	8.9 ± 1.0	9.2 ± 2.1
	Total carotenoids (mg/g)	0.32 ± 0.08	0.87 ± 0.13	0.82 ± 0.19	0.70 ± 0.10
	Extr. glycolipids (g/L)	0.10 ± 0.01	0.01 ± 0.00	0.03 ± 0.02	0.03 ± 0.01
	Beta-glucan (% of DCW)	-	-	-	-
**Mannose**	Total lipids (% of DCW)	37.0 ± 2.3	32.4 ± 2.2	55.8 ± 3.8	51.1 ± 4.1
	Total carotenoids (mg/g)	1.01 ± 0.18	1.02 ± 0.20	1.13 ± 0.22	1.39 ± 0.19
	Extr. glycolipids (g/L)	0.51 ± 0.02	0.61 ± 0.05	1.06 ± 0.08	1.42 ± 0.10
	Beta-glucan (% of DCW)	13.75 ± 0.99	10.59 ± 0.76	10.28 ± 1.83	9.87 ± 0.67
**Glycerol**	Total lipids (% of DCW)	14.9 ± 3.6	16.8 ± 2.8	15.5 ± 1.4	17.2 ± 2.3
	Total carotenoids (mg/g)	0.97 ± 0.11	1.14 ± 0.23	1.18 ± 0.25	1.21 ± 0.18
	Extr. glycolipids (g/L)	0.14 ± 0.00	0.09 ± 0.01	0.23 ± 0.02	0.10 ± 0.03
	Beta-glucan (% of DCW)	3.83 ± 0.54	4.19 ± 0.67	5.00 ± 0.37	5.37 ± 0.80
**Xylose**	Total lipids (% of DCW)	15.6 ± 0.1	15.7 ± 2.5	19.2 ± 0.9	20.3 ± 2.4
	Total carotenoids (mg/g)	0.97 ± 0.22	1.47 ± 0.32	1.13 ± 0.21	1.52 ± 0.30
	Extr. glycolipids (g/L)	0.10 ± 0.01	0.12 ± 0.02	0.15 ± 0.05	0.03 ± 0.01
	Beta-glucan (% of DCW)	2.96 ± 0.39	8.28 ± 1.13	7.59 ± 0.91	7.04 ± 0.76

**Table 7 microorganisms-09-01280-t007:** Biomass composition of *R. kratochvilovae* CCY 20-2-26 cultured on different carbon sources with potassium nitrate.

Carbon	Metabolite	48 h	72 h	96 h	144 h
**Lactose**	Total lipids (% of DCW)	12.5 ± 0.9	16.9 ± 0.6	18.7 ± 1.1	20.2 ± 0.9
	Total carotenoids (mg/g)	0.53 ± 0.10	0.68 ± 0.11	0.78 ± 0.18	0.40 ± 0.09
	Extr. glycolipids (g/L)	0.50 ± 0.11	0.07 ± 0.02	0.05 ± 0.02	0.02 ± 0.01
	Beta-glucan (% of DCW)	-	-	-	-
**Mannose**	Total lipids (% of DCW)	34.1 ± 2.3	40.0 ± 2.9	51.0 ± 2.5	57.5 ± 3.1
	Total carotenoids (mg/g)	1.10 ± 0.11	1.11 ± 0.15	1.31 ± 0.20	2.21 ± 0.16
	Extr. glycolipids (g/L)	0.83 ± 0.17	1.35 ± 0.24	1.68 ± 0.33	1.99 ± 0.36
	Beta-glucan (% of DCW)	17.13 ± 1.24	15.62 ± 1.67	18.14 ± 0.58	19.46 ± 1.85
**Glycerol**	Total lipids (% of DCW)	17.3 ± 1.9	18.0 ± 1.8	18.1 ± 2.0	26.7 ± 2.1
	Total carotenoids (mg/g)	0.66 ± 0.12	0.90 ± 0.09	1.29 ± 0.14	2.72 ± 0.24
	Extr. glycolipids (g/L)	0.05 ± 0.01	0.06 ± 0.01	0.08 ± 0.02	0.13 ± 0.03
	Beta-glucan (% of DCW)	6.61 ± 0.49	5.36 ± 0.41	3.08 ± 0.67	2.69 ± 0.34
**Xylose**	Total lipids (% of DCW)	13.6 ± 1.9	15.2 ± 1.2	22.4 ± 1.7	34.9 ± 2.2
	Total carotenoids (mg/g)	0.72 ± 0.14	1.12 ± 0.20	1.13 ± 0.11	1.97 ± 0.23
	Extr. glycolipids (g/L)	0.05 ± 0.02	0.22 ± 0.04	0.22 ± 0.05	0.25 ± 0.03
	Beta-glucan (% of DCW)	14.08 ± 1.05	13.58 ± 0.81	11.85 ± 1.45	10.00 ± 1.27

**Table 8 microorganisms-09-01280-t008:** Composition of yeast biomass during growth under different C/N ratio.

C/N Ratio	Metabolite	48 h	96 h	144 h	168 h
**20:1**	Total lipids (% of DCW)	16.2 ± 1.5	16.4 ± 1.5	14.3 ± 4.2	14.2 ± 1.5
	Extr. glycolipids (g/L)	0.10 ± 0.02	0.06 ± 0.01	0.19 ± 0.03	0.16 ± 0.01
	Total carotenoids (mg/g)	1.33 ± 0.10	0.72 ± 0.06	0.90 ± 0.03	0.85 ± 0.08
	Beta-glucan (% of DCW)	10.85 ± 0.68	9.30 ± 0.23	6.54 ± 0.25	7.38 ± 0.63
**40:1**	Total lipids (% of DCW)	33.3 ± 1.32	40.7 ± 1.88	28.3 ± 3.6	26.2 ± 1.3
	Extr. glycolipids (g/L)	0.32 ± 0.06	0.35 ± 0.04	0.46 ± 0.02	0.48 ± 0.05
	Total carotenoids (mg/g)	1.07 ± 0.12	0.90 ± 0.20	1.10 ± 0.08	1.03 ± 0.11
	Beta-glucan (% of DCW)	18.45 ± 1.26	15.48 ± 1.41	13.33 ± 1.19	12.83 ± 1.23
**100:1**	Total lipids (% of DCW)	40.1 ± 2.8	53.5 ± 2.9	61.3 ± 3.1	62.1 ± 2.5
	Extr. glycolipids (g/L)	0.95 ± 0.11	1.35 ± 0.15	2.21 ± 0.06	2.25 ± 0.13
	Total carotenoids (mg/g)	1.08 ± 0.14	0.76 ± 0.12	1.48 ± 0.21	1.57 ± 0.22
	Beta-glucan (% of DCW)	19.75 ± 0.91	14.10 ± 1.33	12.04 ± 0.42	10.25 ± 0.98
**120:1**	Total lipids (% of DCW)	42.2 ± 3.86	64.5 ± 4.32	65.3 ± 3.20	66.7 ± 1.50
	Extr. glycolipids (g/L)	0.9 ± 0.03	1.40 ± 0.05	2.19 ± 0.06	2.42 ± 0.08
	Total carotenoids (mg/g)	1.41 ± 0.15	0.76 ± 0.12	1.21 ± 0.09	1.35 ± 0.11
	Beta-glucan (% of DCW)	20.75 ± 0.85	12.08 ± 1.49	14.76 ± 0.95	11.33 ± 1.34

## Data Availability

Publicly available datasets were analyzed in this study. Data of this study are partially available in a publicly accessible repository—PhD thesis of Dana Byrtusova, Dr. This data can be found here: https://www.vutbr.cz/studenti/zav-prace/detail/129244.

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
