# Peer review of "Rhodotorula kratochvilovae CCY 20-2-26—The Source of Multifunctional Metabolites"

_microorganisms, 2021, doi:10.3390/microorganisms9061280_

Round 1
Reviewer 1 Report
Comments on m/s 1223003
Byrtusová et al performed a significant number of fermentations on media composed of glucose, glycerol, xylose, mannose, etc and yeast extract, urea, KNO3, etc, implicated as carbon and nitrogen sources respectively, by a new wt red yeast strain namely Rhodotorula kratochvilovae CCY 20-2-26. The trials were performed in media presenting combinations of carbon and nitrogen sources used, whereas various important yeast compounds (i.e. yeast total biomass, extra-cellular glycolipids, intra-cellular lipids, carotenoids, intra-cellular glycans like α and β) have been produced. The topic of the submitted m/s is certainly of importance. On the other hand, much more elements, considerations, additions and discussions are requested for this m/s to become ready for review. Specifically, the approaches dealing with the synthesis of intra-cellular polysaccharides, their intra-play with the synthesis of intra-cellular lipids, etc, have not at all been considered for discussion, although the last years there is a number of scientific works that have appeared dealing with the production of intra-cellular polysaccharides by oleaginous yeasts when cultured on several types of hydrophilic compounds similar to the ones used in the current study (i.e. lactose, xylose, glycerol, glucose, etc). Likewise, in several instances red yeasts have been employed as cell factories for this purpose, with equally no discussions having appeared on this topic. Specific comments:
1) In all tables and figures that present growth results (i.e. Fig. 1, 2, etc, T. 5, 6, etc), in the relevant fermentation points we need to have the values of the quantity of substrate (glucose, glycerol, mannose, etc) consumed (in g/L). Thus, we can have a quite better idea of the values of total biomass, total lipid and total glycans produced per unit of substrate (glucose, glycerol, mannose, etc) assimilated.
2) I would appreciate to see in separate Tables the fatty acid (FA) composition of cellular lipids produced by Rhodotorula kratochvilovae cultivated on various carbon and nitrogen sources employed.
3) In T. 6, 7, 8, etc theoretically we see only the quantity of β-glycans produced per unit of DCW. Which is the quantity of total glycans? Why do we not see this quantity in the relevant tables? A separate production of glycans (in g/L) reported in the various fermentation configurations of the present study would be highly appreciated.
4) In most cases, the kinetics of glycans (% in DCW as function of culture time; see specifically T. 5 and T. 8) seems to present relatively elevated polysaccharides values at the very early stages of the fermentation (in many cases these values are >20% w/w) and these values drop as the fermentation proceeds simultaneously with the increase of the values of cellular lipid (% in DCW). This is a typical inter-play in the biosynthesis of the cellular polysaccharides and cellular lipids that has been reported for several oleaginous yeast species and mostly Cryptococcus curvatus cultivated on various carbon sources [i.e. lactose, xylose, sucrose, etc – see: Eur J Lipid Sci Technol (2015) 117: 657–672; Biochem Eng J (2020) 162: 107706]. This inter-play has also been reported for “red” yeasts cultivated on glycerol [J Appl Microbiol (2015) 118: 911–927; FEMS Microbiol Lett (2020) fnaa063; Carbon Resour Convers (2021) 4: 61–75] or yeasts of the species Yarrowia lipolytica [Appl Microbiol Biotechnol (2019) 103: 8585–8596; Carbon Resour Convers (2021) 4: 61–75]. In contrast, other yeasts cultivated in nitrogen-limited media (e.g. of the genus Metschnikowia), present a constant increase of total endopolysaccharide values as the fermentation proceeds, and in several cases the quantity of total polysaccharides per DCW presents indeed impressive values (c. 62% w/w – see: FEMS Microbiol Lett (2020) fnaa063). Nothing at all is stated in the discussion of this point (lines 388–395 of the current submission) that is indeed very poorly written and arranged.
5) Separate tables presenting the production of total polysaccharides and lipids (in g/L and % w/w in DCW) of the current submission and their comparisons with the literature would be highly appreciated.
6) Table 7 was cut, and we cannot see it.
7) I do not understand the assay of total glycans (lines 108–111). Total dried biomass has been hydrolyzed with concentrated sulfuric acid. Thereafter, the enzymes exo-β-(1→3)-D-glucanase and β-glucosidase or amyloglucosidase and invertase what exactly do they hydrolyze?
All the above-mentioned points need to be clearly and precisely responded. Literature search should absolutely be made more correctly and professionally. The submitted m/s is not ready for review. The decision, hence, is re-submission after radical revisions, as indicated in the current report.
Author Response
Specific comments:
- In all tables and figures that present growth results (i.e. Fig. 1, 2, etc, T. 5, 6, etc), in the relevant fermentation points we need to have the values of the quantity of substrate (glucose, glycerol, mannose, etc) consumed (in g/L). Thus, we can have a quite better idea of the values of total biomass, total lipid and total glycans produced per unit of substrate (glucose, glycerol, mannose, etc) assimilated
REPLY: Individual sugar consumption was added to the main manuscript as Figure 3A,B, more detailed data are introduced in Suplementary file - Tables S2, S3. Glucose consumption at different nitrogen sources was added to Supplementary file as Table S1.
- I would appreciate to see in separate Tables the fatty acid (FA) composition of cellular lipids produced by Rhodotorula kratochvilovaecultivated on various carbon and nitrogen sources employed.
REPLY: Some of these data were published previously (Ref. 11; 33), more detailed data were added to Supplemetary File as Figures S1, S3, S4.
- In T. 6, 7, 8, etc theoretically we see only the quantity of β-glycans produced per unit of DCW. Which is the quantity of total glycans? Why do we not see this quantity in the relevant tables? A separate production of glycans (in g/L) reported in the various fermentation configurations of the present study would be highly appreciated.
REPLY: Alpha-glucans are produced in most of cells in the range of about 1 – 2% of total glucans. Thus, in the main manuscript we have evaluated only beta-glucans. Complete data of beta-glucans, alpha-glucans and total glucans were added to Supplementary File in the Tables S4, S5, S6. Short comment was added also into the text.
- In most cases, the kinetics of glycans (% in DCW as function of culture time; see specifically T. 5 and T. 8) seems to present relatively elevated polysaccharides values at the very early stages of the fermentation (in many cases these values are >20% w/w) and these values drop as the fermentation proceeds simultaneously with the increase of the values of cellular lipid (% in DCW). This is a typical inter-play in the biosynthesis of the cellular polysaccharides and cellular lipids that has been reported for several oleaginous yeast species and mostly Cryptococcus curvatuscultivated on various carbon sources [i.e. lactose, xylose, sucrose, etc – see: Eur J Lipid Sci Technol (2015) 117: 657–672; Biochem Eng J (2020) 162: 107706]. This inter-play has also been reported for “red” yeasts cultivated on glycerol [J Appl Microbiol (2015) 118: 911–927; FEMS Microbiol Lett (2020) fnaa063; Carbon Resour Convers (2021) 4: 61–75] or yeasts of the species Yarrowia lipolytica [Appl Microbiol Biotechnol (2019) 103: 8585–8596; Carbon Resour Convers (2021) 4: 61–75]. In contrast, other yeasts cultivated in nitrogen-limited media (e.g. of the genus Metschnikowia), present a constant increase of total endopolysaccharide values as the fermentation proceeds, and in several cases the quantity of total polysaccharides per DCW presents indeed impressive values (c. 62% w/w – see: FEMS Microbiol Lett (2020) fnaa063). Nothing at all is stated in the discussion of this point (lines 388–395 of the current submission) that is indeed very poorly written and arranged.
REPLY: Discussion was substantially enlarged and above mentioned works were discussed and compared with the results obtained in present study. Some references were added into List of references.
- Separate tables presenting the production of total polysaccharides and lipids (in g/L and % w/w in DCW) of the current submission and their comparisons with the literature would be highly appreciated.
REPLY: Some of these results were published in our previous study (ref. 24), thus, separate tables were not added. Discussion was enlarged and our findings were discussed and compared with our previous data as well as with the results of other authors.
- Table 7 was cut, and we cannot see it.
REPLY: Table 7 is included into manuscript (highlighted)
7) I do not understand the assay of total glycans (lines 108–111). Total dried biomass has been hydrolyzed with concentrated sulfuric acid. Thereafter, the enzymes exo-β-(1→3)-D-glucanase and β-glucosidase or amyloglucosidase and invertase what exactly do they hydrolyze?
REPLY: Explanation is added to the Methods.
All the above-mentioned points need to be clearly and precisely responded. Literature search should absolutely be made more correctly and professionally. The submitted m/s is not ready for review. The decision, hence, is re-submission after radical revisions, as indicated in the current report.
REPLY: List of references was revised, corrected, and enlarged.

Reviewer 2 Report
Review of the article ‘Rhodotorula kratochvilovae CCY 20-2-26 – the source of multi-variate metabolites’ The manuscript is well written. The results are comprehensive and will be very useful for yeast researches so I recommend the present manuscript to publication in Miroorganisms after minor revision.
Line 74 ‘Two biological replicates were prepared for each strain’ - I do not understand, one yeast strain was examined in the article
Line 93 ‘The cells were washed with sterile water and resuspended in fresh…’ - please give the number of cells
Line - Why were such low (110 rpm) shaking parameters used?
Line 154 – ‘Contents of individual pigments were calculated according to calibration strandards (beta-carotene, astaxanthin, lycopene, torulene, torularhodine) from regress equations’ - from which companies were the standards purchased?
Expand the discussion on the effects of C/N on lipids and carotenoids in terms of bichemistry.
The methodology stated that the content of beta-carotene, torulene, etc. was determined (line 154). Please provide and discuss these results.
Author Response
Review of the article ‘Rhodotorula kratochvilovae CCY 20-2-26 – the source of multi-variate metabolites’ The manuscript is well written. The results are comprehensive and will be very useful for yeast researches so I recommend the present manuscript to publication in Miroorganisms after minor revision.
Line 74 ‘Two biological replicates were prepared for each strain’ - I do not understand, one yeast strain was examined in the article
REPLY: corrected
Line 93 ‘The cells were washed with sterile water and resuspended in fresh…’ - please give the number of cells
REPLY: Corrected
Line - Why were such low (110 rpm) shaking parameters used?
REPLY: Explained in methods; construction of shaker did not allow higher rpm value.
Line 154 – ‘Contents of individual pigments were calculated according to calibration strandards (beta-carotene, astaxanthin, lycopene, torulene, torularhodine) from regress equations’ - from which companies were the standards purchased?
REPLY: Added to the Methods.
Expand the discussion on the effects of C/N on lipids and carotenoids in terms of bichemistry.
REPLY: Done – see above.
The methodology stated that the content of beta-carotene, torulene, etc. was determined (line 154). Please provide and discuss these results.
REPLY: Added in the form of Figures S2, S5, S6, S8 (Supplementary File) and discussed in the text.

Round 2
Reviewer 1 Report
I have appreciated the work performed in the revision of this paper, that now can be accepted.